# Therapeutic Potential of a Novel α_v_β_3_ Antagonist to Hamper the Aggressiveness of Mesenchymal Triple Negative Breast Cancer Sub-Type

**DOI:** 10.3390/cancers11020139

**Published:** 2019-01-24

**Authors:** Billy Samuel Hill, Annachiara Sarnella, Domenica Capasso, Daniela Comegna, Annarita Del Gatto, Matteo Gramanzini, Sandra Albanese, Michele Saviano, Laura Zaccaro, Antonella Zannetti

**Affiliations:** 1Istituto di Biostrutture e Bioimmagini-CNR, 80145 Naples, Italy; billy.hill@ibb.cnr.it (B.S.H.); achiara.sarnella@gmail.com (A.S.); daniela.comegna@unina.it (D.C.); annarita.delgatto@unina.it (A.D.G.); matteo.gramanzini@ibb.cnr.it (M.G.); sandra.albanese@ibb.cnr.it (S.A.); lzaccaro@unina.it (L.Z.); 2Dipartimento di Farmacia, Università degli Studi di Napoli “Federico II”, 80131 Naples, Italy; domenica.capasso@unina.it; 3Istituto di Cristallografia-CNR, 70126 Bari, Italy; msaviano@unina.it

**Keywords:** triple-negative breast cancer, α_v_β_3_ integrin, ψRGDechi, cell migration and invasion, epithelial-mesenchymal transition, stemness

## Abstract

The mesenchymal sub-type of triple negative breast cancer (MES-TNBC) has a highly aggressive behavior and worse prognosis, due to its invasive and stem-like features, that correlate with metastatic dissemination and resistance to therapies. Furthermore, MES-TNBC is characterized by the expression of molecular markers related to the epithelial-to-mesenchymal transition (EMT) program and cancer stem cells (CSCs). The altered expression of α_v_β_3_ integrin has been well established as a driver of cancer progression, stemness, and metastasis. Here, we showed that the high levels of α_v_β_3_ are associated with MES-TNBC and therefore exploited the possibility to target this integrin to reduce the aggressiveness of this carcinoma. To this aim, MES-TNBC cells were treated with a novel peptide, named ψRGDechi, that we recently developed and characterized for its ability to selectively bind and inhibit α_v_β_3_ integrin. Notably, ψRGDechi was able to hamper adhesion, migration, and invasion of MES-TNBC cells, as well as the capability of these cells to form vascular-like structures and mammospheres. In addition, this peptide reversed EMT program inhibits mesenchymal markers. These findings show that targeting α_v_β_3_ integrin by ψRGDechi, it is possible to inhibit some of the malignant properties of MES-TNBC phenotype.

## 1. Introduction

Triple negative breast cancer (TNBC) is a heterogeneous disease, and even though it occurs in only 15–20% of all patients with breast cancer, it is characterized by an extremely high rate of mortality due to metastatic and drug-resistance recurrent disease [1]. The absence of the expression of estrogen receptor, progesterone receptor, and human epidermal growth factor 2 (HER2) in all TNBC sub-types makes them not eligible for targeted therapy. Therefore, many studies are underway to find actionable molecular target to treat patients of this disease. Several attempts have been made to establish a molecular classification of TNBC based on omics approaches. These efforts allowed the identification of different sub-groups, showing a distinct molecular profile with different responses to therapies [2].

Lehmann et al. [2] identified six TNBC sub-types displaying unique gene expression (GE) and ontologies, including two basal-like (BL1 and BL2), an immunomodulatory (IM), a mesenchymal (M), a mesenchymal stem-like (MSL), and a luminal androgen receptor (LAR) sub-type. Recently, this classification was refined from these six subgroups into four (BL1, BL2, M, and LAR), because IM and MSL sub-types represent tumors with substantial infiltrating lymphocytes, and tumor-associated stromal cells, respectively [3]. Similarly, Burstein classified TNBC into four sub-types: Basal-like immunosuppressed (BLIS), basal-like immune-activated (BLIA), mesenchymal (MES), and luminal androgen receptor (LAR) through genetic profiles of 198 TNBC tumors [4]. Mesenchymal TNBC group comprises approximately 30% of TNBCs [2,5] and has been associated with a worse prognosis [2,5,6]. Furthermore, numerous pre-clinical and clinical studies reported that MES-TNBCs are enriched in epithelial-to-mesenchymal transition (EMT) markers [1]. EMT program causes the differentiation of tumor cells from epithelial to mesenchymal phenotypes, resulting in the loss of E-cadherin expression and the acquisition of mesenchymal markers, such as N-cadherin and Vimentin [7]. Furthermore, many evidences suggest that the aggressive nature of MES-TNBC, which correlate with the presence of cancer stem cells (CSCs)show a unique capability for self-renewal, tumor initiation, as well as resistance to traditional cancer therapies [8]. Nowadays, the main biomarkers, that are able to identify CSC sub-population in TNBC, are CD44^+^/CD24^−^ and ALDH1^+^ [8]. Importantly, it has been reported that, in several carcinomas, the EMT program and CSC state are strictly connected to each other and associated with treatment failure, metastases, and cancer relapse [7]. Another characteristic of MES-TNBC sub-type is its ability to form vascular-like networks, a phenomenon described as vascular mimicry (VM) [9] and known to drive metastatic dissemination [10]. Recently, many studies investigated the key role played by integrins in many of the mechanisms underlying the aggressive nature of cancer cells, as well as in the support of tumor micro-environment [11]. Among these cell adhesion receptors, the altered expression of α_v_β_3_ integrin has been correlated with highly malignant tumors, including melanoma [12], glioblastoma [13], breast [14], pancreatic [15], and prostate [16] carcinomas. Most importantly, it the involvement of α_v_β_3_ integrin has been observed in promoting EMT program [17,18,19], stemness [20,21,22], and the resistance of cancer cells to erlotinib [23] and cisplatin treatments [17]. Integrin α_v_β_3_ recognizes the Arg-Gly-Asp motif (RGD) on several extracellular matrix (ECM) proteins [11]. Many ligands containing this sequence are proposed as diagnostic and therapeutic agents in cancer, even if they are not specific for α_v_β_3_ integrin [24]. In our previous studies, we developed and characterized a novel bi-functional chimeric RGD peptide, including a cyclic RGD pentapeptide covalently linked by a spacer to an echistatin domain (RGDechi) that showed a high selectivity for α_v_β_3_ integrin [25,26,27,28,29,30,31]. Recently, this molecule was modified to increase its serum stability, named ψRGDechi [32]. It was tested in vitro and in vivo, for its ability to bind and inhibit α_v_β_3_ integrin function in melanoma [32].

In this study, we found that both αv and β_3_ sub-units are highly expressed in mesenchymal TNBC sub-type. Therefore, we focused on the possibility of hampering some aspects correlating with the aggressive behavior of MES-TNBC cells, such as their ability to migrate, to invade ECM, to form VM structures and mammospheres by blocking α_v_β_3_ integrin with ψRGDechi peptide. In addition, we investigated whether the treatment of MES-TNBC cells with this novel peptide could affect the signaling pathway involved in EMT program.

## 2. Results

### 2.1. α_v_β_3_ Integrin Expression Is Associated with MES-TNBC Mesenchymal Sub-Type

First, to explore whether the expression of integrin α_v_β_3_ could be associated with TNBC, we analyzed the gene expression of both sub-units, αv (ITGAV) and β_3_ (ITGB3), in a public data-set of 198 TNBC samples, specifically characterized and divided according to Burstein [4] into four sub-types, including luminal-androgen receptor (LAR), mesenchymal (MES), basal-like immunosuppressed (BLIS), and basal-like immune-activated (BLIA). We found that the expression of both sub-units, αv and β_3,_ is highest in the mesenchymal sub-group with respect to the other TNBC sub-types (Figure 1A).

### 2.2. Expression of α_v_β_3_ Integrin in MES-TNBC Cell Lines

We evaluated the expression of α_v_β_3_ integrin in two MES-TNBC cell lines, MDA-MB-231, and BT-549, by flow cytometry. As shown in Figure 1B, we observed that both cell lines express very high levels of α_v_β_3_ with a mean fluorescence intensity (MFI) of 103.38 for MDA-MB-231 and 83.98 for BT-549, respectively.

### 2.3. ψRGDechi Inhibits MES-TNBC Cell Adhesion

Given the crucial role played by α_v_β_3_ integrin on cell adhesion to the extracellular matrix, we tested the effect of ψRGDechi on the ability of MDA-MB-231 and BT-549 TNBC cells to adhere to vitronectin. MES-TNBC cells were treated with different concentrations of ψRGDechi for 30 min and then seeded on plates coated with vitronectin. As shown in Figure 2, ψRGDechi significantly inhibited cell adhesion in a concentration-dependent manner, starting at 5 µM in both MES-TNBC cell lines. A similar effect was observed after treatment with anti-α_v_β_3_ antibody LM609 (10 μg/mL) (Figure 2), whereas cell incubation with scrambled peptide had no affect on MES-TNBC cell adhesion.

### 2.4. ψRGDechi Hampers MES-TNBC Cell Migration

Recently, we reported on the strong ability of MES-TNBC cells to migrate and invade, and to form metastases in vivo [33,34]. Therefore, we investigated whether ψRGDechi targeting α_v_β_3_ could interfere with these mechanisms in BT-549 and MDA-MB-231 cell lines. These cells were treated in serum-free medium containing different concentrations of ψRGDechi (from 1 to 50 µM), scrambled-peptide (50 µM) and anti-α_v_β_3_ antibody (10 µg/mL), and seeded on the upper compartment of the Boyden chamber, whereas 1% and 10% FBS were added to the lower compartment and used as chemo-attractants. A significant reduction of cell migration was observed in BT-549 cells treated with ψRGDechi at 10 µM (*p* < 0.001) and 50 µM (*p* < 0.0001), with respect to untreated (10% FBS) and scrambled-peptide treated cells, whereas MDA-MB-231 cells showed a significant delay of migration after treatment with ψRGDechi already at 1 µM (*p* < 0.01) (Figure 3A). Anti-α_v_β_3_ antibody caused a strong inhibition of migration in both cell lines as expected. In addition, to confirm the ability of ψRGDechi to hamper MES-TNBC cell migration, we performed in vitro wound healing assay. Monolayers of MDA-MB-231 and BT-549 cells were scratched and images were taken at 0, 24, and 48 h after wounding. When MES-TNBC cell lines were grown in the presence of 10 µM ψRGDechi, the wound healing was significantly delayed compared to untreated cells (10% FBS) at 24 h (MDA-MB-231, *p* < 0.01; BT-549, *p* < 0.01) and at 48 h (MDA-MB-231, *p* < 0.01; BT-549, *p* < 0.001) (Figure 3B). As expected, anti-α_v_β_3_ decreased wound closure whereas scrambled-peptide did not. In addition, we observed that ψRGDechi (from 1 to 50 µM) had no effect on cell proliferation at 24, 48 and 72 h as assessed by MTS assay (Appendix A).

### 2.5. ψRGDechi Hampers MES-TNBC Cell Invasion

The crucial contribution given by α_v_β_3_ integrin in the different steps of cancer cell dissemination and organ-specific metastasis formation [11,32] is well-known. Therefore, we investigated the role played by α_v_β_3_ in MES-TNBC cell invasion and tested the effect of ψRGDechi on MDA-MB-231 and BT-549 cells, using trans-well chambers with matrigel coated membranes to mimic extracellular matrix. As shown in Figure 4, we observed a significant decrease in tumor cell invasiveness when MES-TNBC cells were treated with 10 μM ψRGDechi or 10 µg/mL α_v_β_3,_ in comparison with untreated and scrambled-peptide treated cells. In particular, ψRGDechi was able to significantly inhibit tumor cell invasion of 92% in BT-459 (*p* < 0.0001) and of 80% in MDA-MB-231 (*p* < 0.0001), with respect to untreated cells.

### 2.6. ψRGDechi Inhibits Ability of MES-TNBC Cells to form Vascular-Like Structures

Another peculiar characteristic of the MES-TNBC cells is their capability to acquire endothelial-like properties and form vascular-like tubular networks correlating with metastases and poor clinical outcome [35]. To assess the effect of ψRGDechi in interfering with this mechanism, BT-549 and MDA-MB-231 cells were seeded in Matrigel-coated 24-wells in 2% FBS medium in the presence of 10 µM ψRGDechi, 10 µM scrambled-peptide, and 10 µg/mL anti-α_v_β_3_ antibody. In agreement with their remarkable degree of plasticity, both cell lines formed very organized channels at 24 h. Notably, we found that ψRGDechi strongly prevented the vasculogenic capacity of these cells similarly to the anti-α_v_β_3_ antibody, whereas scrambled peptide was not able to affect this phenomenon (Figure 5).

### 2.7. ψRGDechi Inhibits Ability of MES-TNBC Cells to Form Spheroids 

Under anchorage-independent serum-free culture conditions, TNBC cells can form mammospheres that show features of cancer stem cells [36]. Integrin α_v_β_3_ is considered a putative marker of breast, lung, and pancreatic carcinomas with stem-like properties and high resistance to treatments [11]. To investigate whether ψRGDechi could inhibit the formation and the size of mammospheres, cells left untreated or treated with 10 µM ψRGDechi, 10 µM scrambled-peptide and 10 µg/ml anti-α_v_β_3_ antibody were seeded in Ultra-Low attachment plates and allowed to grow for 7 days. As shown in Figure 6, ψRGDechi caused, in both MES-TNBC cell lines, a significant reduction of spheroid number (64% for MDA-MB-231 and 76% for BT-549; *p* < 0.0001) and size (73% for MDA-MB-231 and 70% for BT-549; *p* < 0.001), when compared with untreated and scrambled-peptide treated cells. Anti-α_v_β_3_ antibody behaved in an analogous manner to ψRGDechi.

### 2.8. ψRGDechi Reverses EMT Program in MES-TNBC Cell Lines

Recently, α_v_β_3_ expression has been shown to play a pivotal role in EMT program [19,37] that is widely accepted to be closely associated with many of the mechanisms herein investigated [7]. To explore whether ψRGDechi is able to hamper EMT program in the TNBC cell lines, we analyzed N-cadherin, Slug, Vimentin (mesenchymal markers), and E-cadherin (epithelial marker) expression in MDA-MB-231 and BT-549 cells treated with 10 µM ψRGDechi and scrambled peptide. Interestingly, we observed that ψRGDechi peptide caused a reduction in Vimentin and Slug levels in both cell lines. Conversely, a decrease of N-cadherin expression was observed only in BT-549 cells, in agreement with other reports [38,39] that did not detect this protein in MDA-MB-231 cells (Figure 7). Probably, this dissimilar expression of mesenchymal markers among these cell lines could be correlated with their different sensitivity for ψRGDechi in the processes here analyzed. Notably, the inhibition of α_v_β_3_ also enhanced E-cadherin levels, thus reversing the mesenchymal phenotype of TNBC cells. In addition, we found that ψRGDechi reduced the expression of p-AKT, one of the main pathway activated by α_v_β_3_ and involved in EMT program [17,19].

## 3. Discussion

Despite optimal systemic chemotherapy, fewer than 30% of women with metastatic breast cancer survive five years after diagnosis, and virtually all women with metastatic TNBC will ultimately die of their disease [40]. Indeed, many TNBC patients that initially respond to conventional therapies subsequently develop drug resistance, leading to disease relapse [1]. Recent studies reported that aberrant activation of EMT program and stemness features are strictly associated with the ability of tumor cells to invade, disseminate to distant sites, and survival to treatments [7]. The genetic profile of MES-TNBC sub-group is characterized by altered expression of molecules involved in these mechanisms [3]. As describe by Fedele et al., the main pathways underlying the transition of breast cancer cells to a mesenchymal state may be activated by means of genetic/epigenetic alterations, paracrine stimulation from neighbor cells, or direct interaction with ECM components [41]. Conversely, the mechanistic link between EMT and CSC status remains largely elusive, although it has been hypothesized that the activation of EMT results in the establishment of several autocrine signaling loops, such as TGFβ-SMAD and Wnt-β-catenin pathways, that contribute to the induction and maintenance of stem-cell properties [7].

In addition, MES-TNBC cells for their high plasticity are able to undergo endothelial trans-differentiation forming vessel-like structures [35], which provide a blood supply for tumor growth and promote metastasis with mechanisms distinct from classical angiogenesis [10]. It has been reported that zinc-finger transcription factor slug is capable to promote vasculogenic mimicry in hepatocellular carcinoma by the induction of EMT, pluripotency and CSCs-like phenotype [42].

Notably, recent findings described the trans-membrane vitronectin receptor, integrin α_v_β_3_, as a common mediator in the regulation of EMT [43], CSC [21], and VM [35] in highly aggressive carcinomas. The treatment of TNBC cells with β_3_ integrin siRNA, delivered by ECO-based nanoparticles caused a down-regulation of β_3_ expression attenuated TGFβ-mediated EMT and invasion in vitro and in vivo [37]. Interestingly, IL32 secreted by cancer-associated fibroblasts, binding to β_3_ integrin was able to up-regulate EMT markers and increase TNBC cell invasion [44]. In addition, Luo et al showed that 14, 15-EET enhanced EMT and cisplatin resistance up-regulating the expression of α_v_β_3_ integrin in MDA-MB-231 cells [17]. Many studies reported the involvement of PI3K/AKT pathway to mediate EMT activation by α_v_β_3_ integrin [17,18,19]. The co-expression of α_v_β_3_ and Slug identified rare stem-like cells and were associated with tumor progression in breast cancer [20]. Recently, it has been reported that the expression of integrin α_v_β_3_ in CSCs may confer not only tumorigenic features, but also metastatic capabilities through vitronectin engagement [22]. Furthermore, a key role of integrin α_v_β_3_ as a cancer stem cell driver associated with resistance to EGFR inhibitors has been demonstrated in breast, lung, and pancreatic carcinomas [23]. Recently, we proved that targeting EGFR with a specific aptamer, the matrix-induced interaction of the receptor with integrin α_v_β_3_ on membranes of TNBC cells, was impaired and consequently integrin α_v_β_3_-dependent cell adhesion and VM in 3D cell culture, was inhibited [35,45].

In the present study, we investigated the role of α_v_β_3_ in activating the mechanisms underlying the aggressive behavior of MES-TNBC, and we exploited the possibility of using a new molecule targeting α_v_β_3_ to reduce the metastatic potential of this carcinoma. Unfortunately to date, despite many efforts to find specific biomarkers for improving therapeutic approach of each TNBC sub-group, due to the lack of specific targets, the management of this disease is still challenging. In our previous studies we developed the RGDechi peptide, which was able to bind and specifically recognize α_v_β_3_ integrin receptor without cross-reacting with αvβ5 and αIIbβ3 integrins in vitro and in vivo [25,26]. This peptide was characterized for its ability to inhibit α_v_β_3_ function in glioblastoma [26] and melanoma cells [27,30]. In addition, RGDechi, labeled with Indium-111 and Fluorine-18, was able to visualize selectively α_v_β_3_ over-expressing xenografts by single-photon emission computed tomography (SPECT), and positron emission tomography (PET), respectively [26].

Recently, we designed, synthetized and biologically characterized a novel peptide derived from RGDechi, and namely ψRGDechi, with improved stability in serum, due to a modified amide bond at the main protease cleavage site [32]. Our studies, which were carried out in mice bearing melanoma tumors, demonstrated that ψRGDechi-NIR750 was more efficient in targeting α_v_β_3_ over-expressing xenografts with respect to the unmodified peptide, RGDechi-NIR750 [32].

Here, we report, by the analysis of a public data-set of 198 TNBC, that the mesenchymal sub-group expresses the highest levels of αv and β_3_ sub-units, wtih respect to the other sub-types. We found that by targeting α_v_β_3_ integrin with the selective peptide, ψRGDechi, it is possible not only to hamper adhesion, migration, and invasion of TNBC cells, but also to block their capability to form vascular-like structures and mammospheres. Notably, the EMT program involved in all these mechanisms and strongly activated in TNBC cells, was reversed after their treatment with ψRGDechi, as shown by the reduction of mesenchymal markers (N-cadherin, Vimentin and Slug) and the enhancement of epithelial marker (E-cadherin) (Figure 8).

## 4. Materials and Methods 

### 4.1. In Silico Analysis of the Expression of α_v_β_3_ Subunits in TNBC

For integrin sub-units αV (ITGAV) and β_3_ (ITGB3) mRNA expression analysis and correlation with clinical, molecular, and cell phenotype, the Genomics Analysis and Visualization platform ((Internet) R2: genomics analysis and visualization platform. http://r2.amc.nl.) was used [33]. The analysis was performed with the following data-set: (GSE76124) which includes 198 TNBC tumors from MD Anderson Cancer Center (Houston, TX, USA) [4]. All correlations were assessed by Pearson’s χ^2^ test or one-way analysis of variance (ANOVA), through the R2 platform and presented in Box-Plots and subdivided in TNBC subtypes.

### 4.2. Cell Lines and Culture Conditions

MDA-MB-231 and BT-549 MES-TNBC cell lines, purchased from American Type Culture Collection (ATCC, Manassas, VA, USA) were grown as previously reported [33,34,35].

### 4.3. Analysis of α_v_β_3_ Integrin Expression in MES-TNBC Cells Using Flow Cytometry

TNBC cells grown to a confluency of 80% were harvested with EDTA and re-suspended in PBS. A quantity of 1 × 10^6^ cells were incubated with FITC mouse antibody against human integrin α_v_β_3_ (LM609; Millipore, Burlington, MA, USA). Isotype-matched antibodies were used as controls (Caltag, Burlingame, CA, USA). After 30 min at 4 °C, the cells were washed and analyzed using a FACSCalibur System (BD Biosciences, San Jose, CA, USA) [27].

### 4.4. Cell Adhesion Assay

Cell adhesion was performed as previously reported with some modifications [46,47]. MDA-MB-231 and BT-549 cells (8 × 10^4^ cells/well) were suspended and mixed in a binding solution (Hank’s balanced salt solution: 50 mM HEPES, 1 mg/mL BSA, 1 mM CaCl_2_, 1 mM MgCl_2_, 1 mM MnCl_2_, pH 7.4, Sigma Aldrich, Milano, Italy) with ψRGDechi (from 0.1 to 50 µM) for 30 min at room temperature and then plated on 96-well plates, pre-coated with 5 μg/mL vitronectin and allowed to attach for 2 h. The non-adherent cell were removed using PBS, and attached cells were stained using a 0.1% crystal violet solution in 25% methanol for 30 min. All the results are expressed as the percentage of adherent cells considering the untreated as 100%.

### 4.5. Cell Proliferation Assay

The viability of MDA-MB-231 and BT-549 cells (5.0 × 10^3^ cells/well, 96-well plates), untreated and treated with ψRGDechi (from 1 to 50 µM), at times of 24, 48, and 72 h was assessed as previously reported [48] by CellTiter 96 AQueous One Solution Cell Proliferation Assay (Promega BioSciences Inc., Fitchburg, WI, USA) using 3-(4,5-dimethylthiazol-2yl)-5-(3-carboxymethoxy-phenyl)-2-(4-sulfophenyl)-2H tetrazolium (MTS), and according to the manufacturer’s instructions. After 1-hour incubation with MTS, absorbance was read in a plate reader (Multiskan RC, Thermo Scientific, Waltham, MA, USA) at a wavelength of 490 nm. The data are expressed as percentage of viable cells, considering the untreated control cells as 100%.

### 4.6. Cell Migration Assay

Cell migration was performed as previously reported [49,50] using a 24-well Boyden chambers (Corning, NY, USA) with inserts of polycarbonate membranes (8µm pores). MDA-MB-231 and BT-549 cells (0.5 × 10^5^/well) were re-suspended in 100 µl of serum-free medium in the presence or absence of different concentration of ψRGDechi (50 µM, 10 µM, 1µM), scrambled-peptide (50µM), and blocking anti-α_v_β_3_ antibody LM609 (10 μg/mL) (Millipore, Burlington, MA, USA) and were seeded in the upper chamber. After the addition of 1% FBS or 10% FBS in the lower chamber as chemo-attractants, the trans-well were put for 24 h at 37 °C in a humidified incubator in 5% CO_2_. The not migrated cells were removed with cotton swabs, whereas the cells that had migrated were visualized by staining the membrane with 0,1% crystal violet in 25% methanol. Ten random fields/filter were counted under a phase contrast microscope (Leica, Wetzlar, Germany) and images were captured by a digital camera (Canon, Tokyo, Japan) attached to the microscope. All experiments were performed at least three times and the results are expressed as the percentage of migrating cells, considering the untreated control sample as 100%.

### 4.7. Wound Healing Assay

In vitro wound model was performed using a scratch assay [51,52]. MDA-MB-231 and BT-549 cells grown as confluent monolayers in 6-well plates were scratched with pipette tips to create wounds. After the removal of the detached cells, medium containing 1% FBS, 10% FBS, ψRGDechi (10 µM), scrambled-peptide (10 µM), and blocking anti-α_v_β_3_ antibody LM609 (10 μg/mL), were added to cells, and the plates were incubated at 37 °C in a humidified incubator in 5% CO_2_ for 24 h and 48 h. Each scratch area was photographed at 0, 24, and 48 h. The distance between the edges of the scratch was measured by ImageJ, the average distance was quantified and the extent of wound closure was determined as follows: Wound closure (%) = 1 − (wound width tx/wound width t0) × 100. All experiments were performed at least three times.

### 4.8. Cell Invasion Assay

The invasion assay was performed using the Boyden chamber with membranes (8 µm pores) coated with 50 μL of diluted Matrigel (1:5 in PBS) (BD Biosciences, San Jose, CA, USA). MDA-MB-231 and BT-549 cells (1 × 10^5^/100 μL serum-free medium per well) were harvested, suspended in serum free medium alone or containing ψRGDechi (10 µM), scrambled peptide (10 µM) and anti-α_v_β_3_ antibody (10 µg/mL), and placed in the top chamber. In the lower chamber medium containing 1% FBS or 10% FBS was added and used as chemo-attractant. Cells were allowed 72 h to invade in a humidified incubator with 5% CO_2_ at 37 °C. To visualize and analyze invading cells, the same experimental procedure described above for cell migration assay was performed.

### 4.9. Tube Formation Assay to Measure In Vitro VM of MES-TNBC Cells

An analysis of the ability of MES-TNBC cells to form vascular-like tubular structures was performed as previously described [35]. Harvested MDA-MB-231 (1 × 10^4^ cells) and BT-549 (8 × 10^5^ cells) were suspended in 100 µL medium containing 2% FBS in the presence or absence of ψRGDechi (10 µM), scrambled peptide (10 µM), and anti-α_v_β_3_ antibody (10 µg/mL). Treated cells were then seeded into 24-well plates pre-coated with 80 µL/well Matrigel and incubated at 37 °C and 5% CO_2_ for 24 h. Tube formation was analyzed under a phase-contrast microscopy and the number of tubular structures, that were identified as complete loops, were quantified. Images were taken at 10× and an average of the number of complete loops was calculated from 3–5 random fields by a macro made with ImageJ software (v1.46r, Bethesda, MD, USA).

### 4.10. Spheroid Formation Assay of MES-TNBC Cells

BT-549 and MDA-MB-231 cells (5 × 10^4^/well) were seeded in Ultra-Low attachment 6-multi-well-plates (Corning, NY, USA) and grown in serum-free DMEM supplemented with B27 (1×), bFGF (20 ng/mL) EGF (10 ng/mL). Cells were incubated at 37 °C with 5% CO_2_ for 7 days. Spheroid formation was analyzed under a phase-contrast microscopy, and size and number of formed spheroids was calculated using imageJ.

### 4.11. Cell Lysates Preparation and Western Blot Analysis

Whole-cell lysates and Western blot analysis were performed as previously described [53,54]. Equal amounts of proteins from cells were separated by 4–12% SDS-PAGE and were transferred to a nitro-cellulose membrane. The blots were blocked for 1 hour with 5% non-fat dry milk and then incubated over night with the following primary antibodies: Anti-N-cadherin, anti-Vimentin, anti-Slug (CST-9782; Cell Signaling Technology Inc, Santa Cruz, CA, USA), anti-E-cadherin (ab1416; abcam), anti p-AKT (CST-9271; Cell Signaling Technology Inc), and anti-AKT (CST-9272; Cell Signaling Technology Inc.) anti-Tubulin (sc-5286; Santa Cruz Biothecnology, Santa Cruz, CA, USA) and anti-Actin (A4700; Sigma-Aldrich, St. Louis, MO, USA). After washing with 0.1% Tween-20 in PBS, the filters were incubated with their respective secondary antibodies for 1h and analyzed using the enhanced chemiluminescence (ECL) system. Densitometric analyses were performed on at least two different expositions to assure the linearity of each acquisition using Image J software (v1.46r).

### 4.12. Statistical Analysis

The results were obtained from at least three independent experiments and are expressed as means ± standard deviation. Data were analyzed with GraphPad Prism statistical software 6.0 (GraphPad Software, La Jolla, CA, USA), and the significance was determined using Student’s *t* test. A *p*-value < 0.05 was considered statistically significant.

## 5. Conclusions

In summary, our findings indicate by that targeting α_v_β_3_ integrin in MES-TNBC cells with ψRGDechi, it is possible to reduce their aggressive behavior. However, further in vivo studies are needed in order to validate this novel peptide as a potential therapeutic agent in the treatment of this disease.

## Figures and Tables

**Figure 1 cancers-11-00139-f001:**
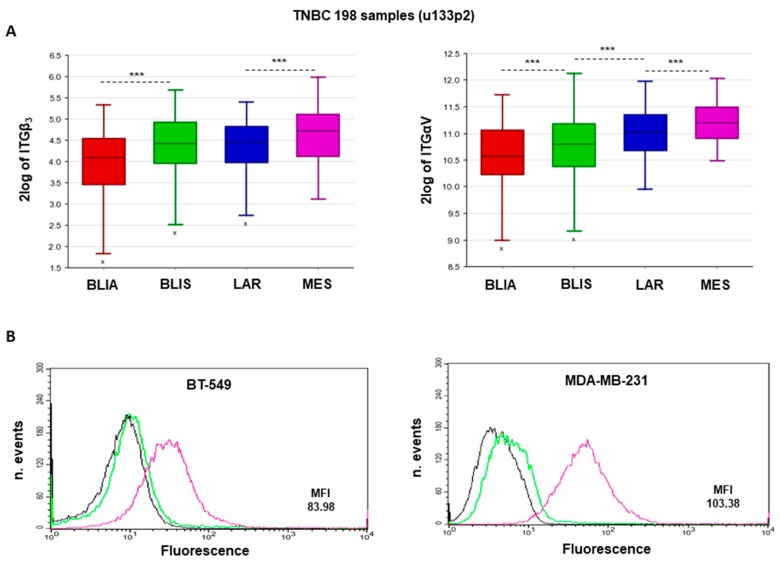
α_v_β_3_ integrin expression is associated with triple-negative breast cancer (TNBC) mesenchymal subtype. (**A**) Analysis of α_V_ (ITGAV) and β_3_ (ITGB3) mRNA expression and their correlation to clinical, molecular, and cell phenotype, was performed on 198 TNBC tumors from MD Anderson Cancer Center (dataset: GSE76124). All correlations were assessed by Pearson’s χ^2^ test or with one-way analysis of variance (ANOVA), through the R2 platform (Academic Medical Center, Netherlands) and presented in BoxPlots and subdivided into TNBC sub-types. BLIA: Basal-like immune-activated; BLIS: Basal-like immune-suppressed; LAR: Luminal-androgen receptor; MES: mesenchymal. *** *p* < 0.0001. (**B**) MES-TNBC cells (1 × 10^6^) were incubated with FITC mouse antibody against human integrin α_v_β_3_ (LM609) and analyzed using a FACSCalibur System (BD Biosciences, San Jose, CA, USA). Isotype-matched antibodies were used as controls.

**Figure 2 cancers-11-00139-f002:**
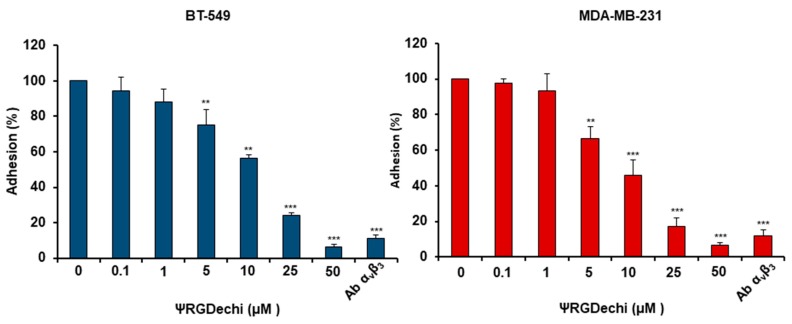
ψRGDechi inhibits MES-TNBC cell adhesion. BT-549 and MDA-MB-231 cells (8 × 10^4^ cells/well) were suspended and mixed in a binding solution with ψRGDechi (from 0.1 to 50 µM) or anti-α_v_β_3_ antibody LM609 (10 μg/mL) (Millipore, Burlington, MA, USA), for 30 min at room temperature, then seeded on plates pre-coated with 5 μg/mL vitronectin and allowed to attach for 2 h. The non-adherent cells were removed using PBS, and the attached cells were stained using a 0.1% crystal violet solution in 25% methanol for 30 minutes. All the results are expressed as the percentage of adherent cells considering the untreated as 100%. Bars depict mean ± SD of three independent experiments. *** *p* < 0.0001; ** *p* < 0.001.

**Figure 3 cancers-11-00139-f003:**
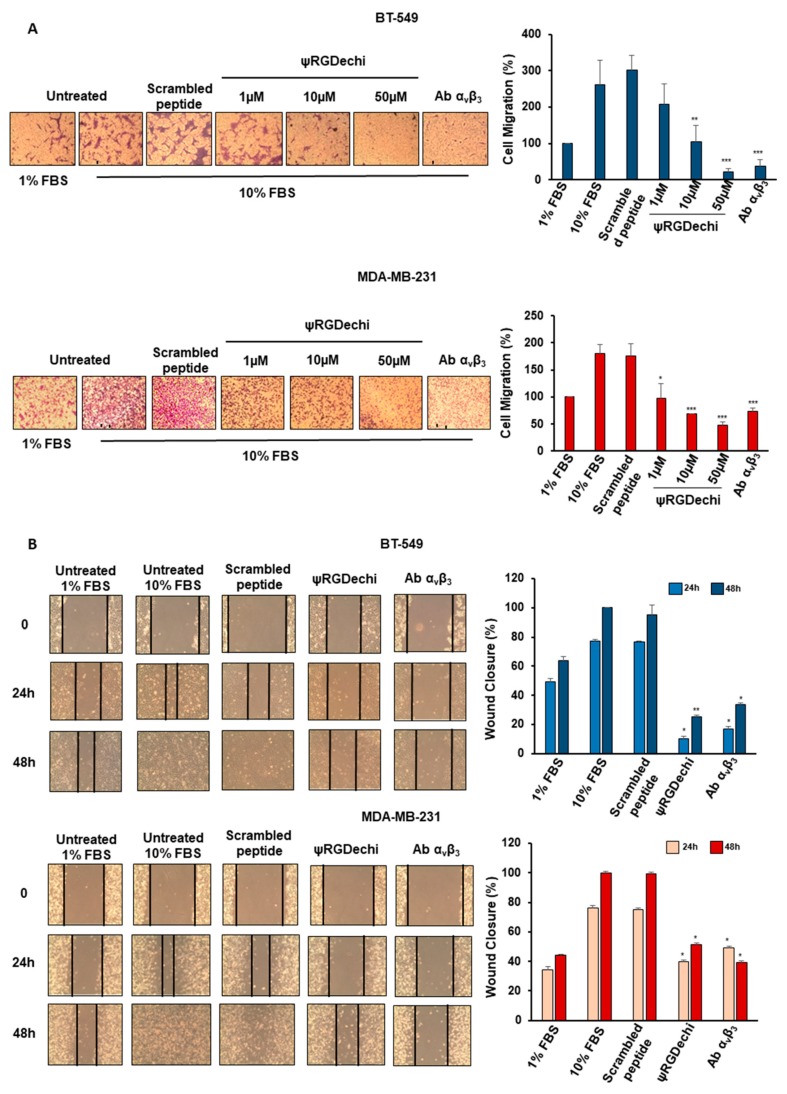
ψRGDechi inhibits MES-TNBC cell migration and cell wound healing ability. (**A**) Cell migration was performed using a 24-well Boyden chamber. BT-549 and MDA-MB-231 cells (0.5 × 10^5^ cells/well) were re-suspended in 100 µL of serum-free medium in presence or absence of different concentration of ψRGDechi (50 µM, 10 µM, 1 µM), scrambled-peptide (50 µM), and blocking anti-α_v_β_3_ antibody LM609 (10 μg/mL) and seeded in the upper chamber. Medium containing 1% FBS or 10% FBS was added to the lower chamber as a chemoattractant. The results are expressed as the percentage of migrating cells considering the untreated control sample (1% FBS) as 100%. **B**) Both cell lines were grown as confluent monolayers in 6-well plates were scratched with pipette tips to create wounds. After removal of detached cells, medium containing 1% FBS, 10% FBS, ψRGDechi (10 µM), scrambled-peptide (10 µM) and blocking anti-α_v_β_3_ antibody LM609 (10 μg/mL), were added to cells. Each scratch area was photographed at 0, 24, and 48 h. The distance between the edges of the scratch was measured by ImageJ, the average distance was quantified and the extent of wound closure was determined as follows: wound closure (%) = 1 − (wound width tx/wound width t0) × 100. Bars depict mean ± SD of three independent experiments. *** *p* < 0.0001; ** *p* < 0.001; * *p* < 0.01.

**Figure 4 cancers-11-00139-f004:**
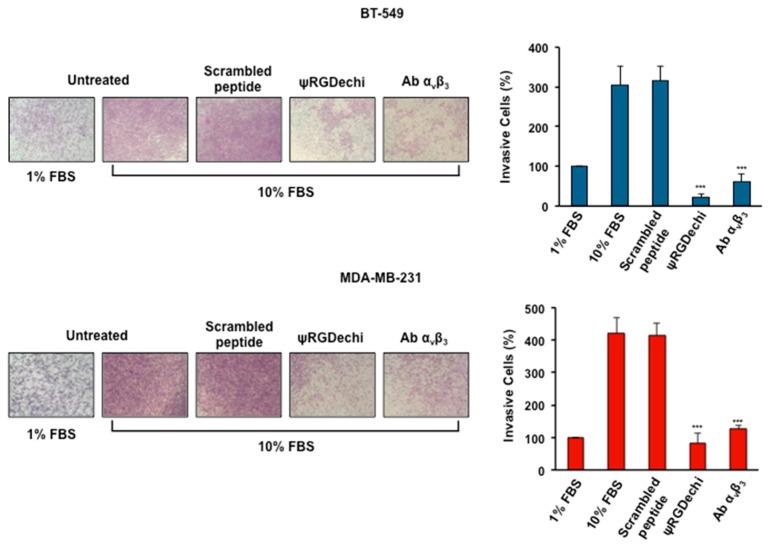
ψRGDechi inhibits MES-TNBC cell ability to invade extracellular matrix. The invasion assay was performed using the Boyden chamber with membranes (8 µm pores) coated with Matrigel. BT-549 and MDA-MB-231 (1 × 10^5^) cells were harvested, suspended in serum free medium alone or containing, ψRGDechi (10 µM), scrambled peptide (10 µM) and anti-α_v_β_3_ antibody (10 µg/mL) and placed in the top chamber. In the lower chamber medium containing 1% FBS or 10% FBS was added and used as a chemoattractant. All experiments were performed at least three times and the results are expressed as the percentage of invasive cells considering the untreated control sample (1% FBS) as 100%. Bars depict mean ± SD of three independent experiments. *** *p* < 0.0001.

**Figure 5 cancers-11-00139-f005:**
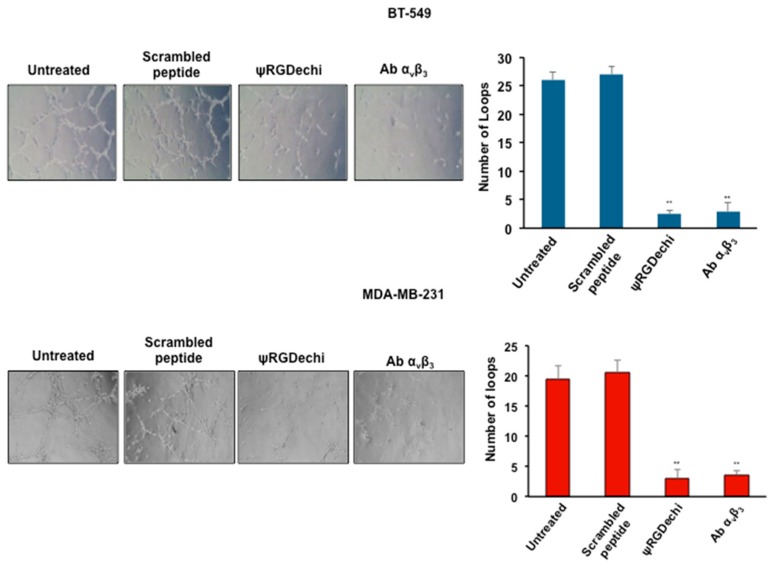
ψRGDechi inhibits MES-TNBC cell ability to form vascular-like structures. Harvested MDA-MB-231 (1 × 10^5^) and BT-549 (8 × 10^4^) cells were suspended in 100 µL medium containing 2% FBS in presence or absence of ψRGDechi (10 µM), scrambled peptide (10 µM) and anti-α_v_β_3_ antibody (10 µg/mL). Treated cells were then seeded into 24-well plates pre-coated with 80 µL/well Matrigel and incubated at 37 °C and 5% CO_2_ for 24 h. Representative images were taken at 10× and average of the number of complete loops was calculated from 3–5 random fields by a macro made with ImageJ software. Bars depict mean ± SD of three independent experiments. ** *p* < 0.001.

**Figure 6 cancers-11-00139-f006:**
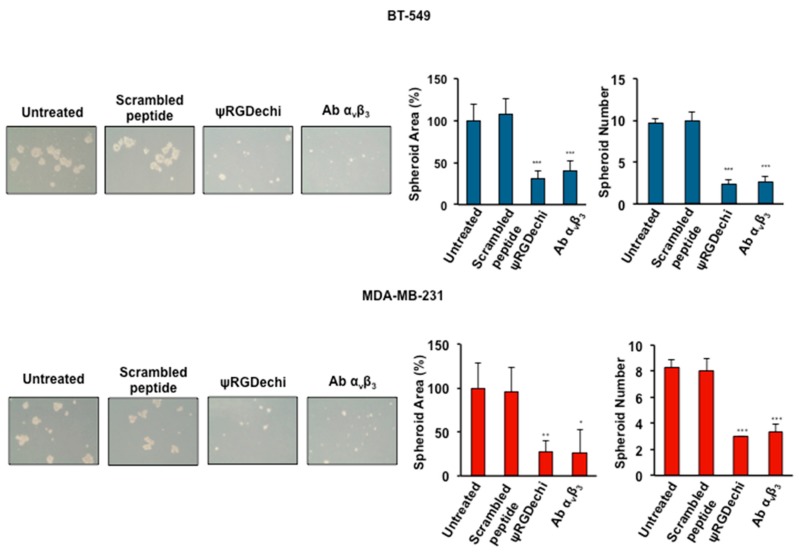
ψRGDechi inhibits MES-TNBC cell ability to form spheroids. BT-549 and MDA-MB-231 cells (5 × 10^4^/well) were seeded in Ultra-Low attachment 6-multiwell-plates and grown in serum-free DMEM supplemented with B27, bFGF (20 ng/mL) EGF (10 ng/mL). Cells were incubated at 37 °C with 5% CO_2_ for 7 days. Spheroid formation was analyzed under a phase-contrast microscopy and size and number of formed spheroids was calculated using imageJ. Bars depict mean ± SD of three independent experiments. *** *p* < 0.0001; ** *p* < 0.001; * *p* < 0.01.

**Figure 7 cancers-11-00139-f007:**
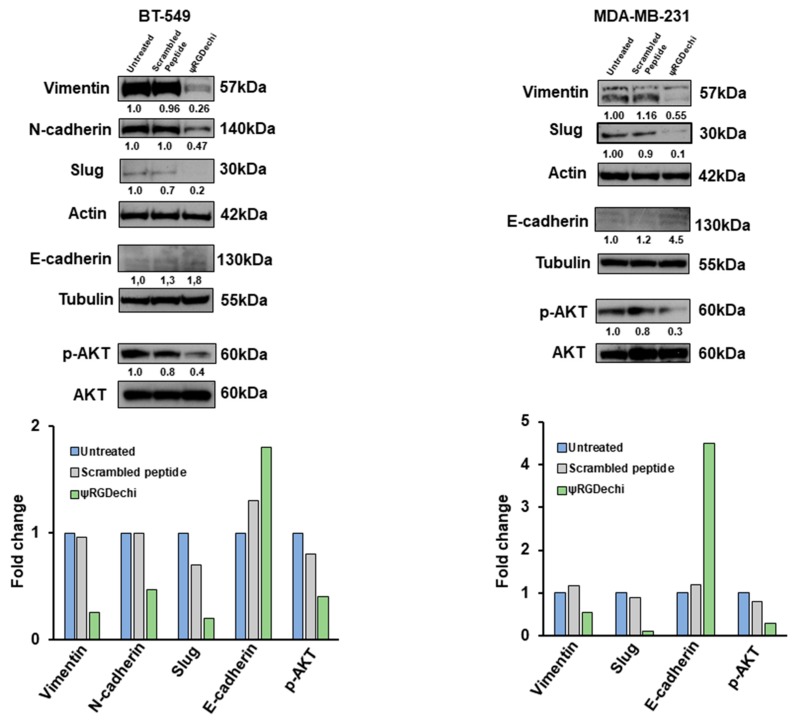
ψRGDechi inhibits EMT program in MES-TNBC cells. Analysis by western blot of Vimentin, N-cadherin, Slug, E-cadherin, p-AKT and AKT levels in untreated, scramble-peptide treated and ψRGDechi treated BT-549 and MDA-MB-231 cells. Actin and Tubulin were used as loading control. Values below the blot indicate signal levels relative to the untreated cells, which were arbitrarily set to 1. Representative data from one of three experiments are shown.

**Figure 8 cancers-11-00139-f008:**
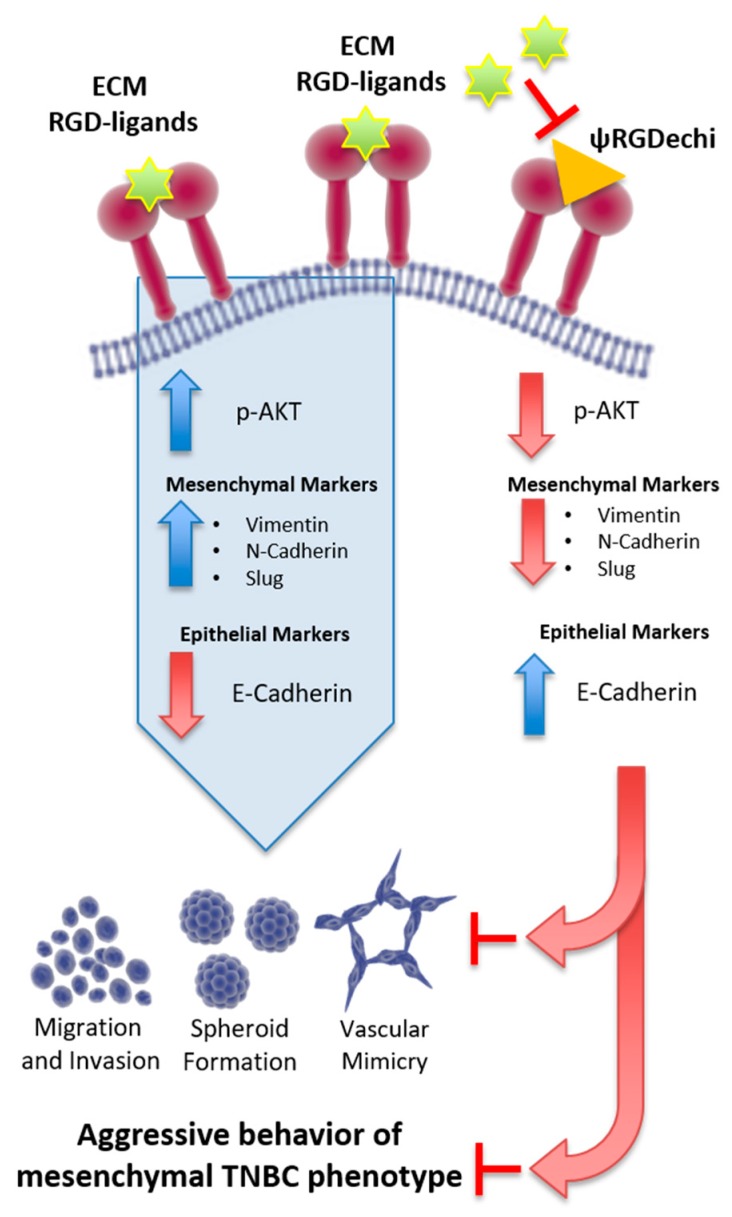
Schematic model illustrating that the blocking of α_v_β_3_ integrin by ψRGDechi reverses the EMT program and reduces migration, invasion, vascular mimicry and stemness in MES-TNBC.

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
