# Peer review of "Therapeutic Potential of a Novel α_v_β_3_ Antagonist to Hamper the Aggressiveness of Mesenchymal Triple Negative Breast Cancer Sub-Type"

_cancers, 2019, doi:10.3390/cancers11020139_

Round 1
Reviewer 1 Report
Hill et al. inhibited integrin αvβ3 with ψRGDechi in mesenchymal subtype of triple-negative breast cancer (TNBC). The authors found higher integrin αv and β3 expressions in mesenchymal subtype of TNBC than the other TNBC subtypes. Treatment with ψRGDechi reduced adhesion, migration, invasion, formation of vascular-like structure and anchorage-independent growth in mesenchymal TNBC cell lines. Inhibition of αvβ3 with ψRGDechi reduced the expression levels of mesenchymal markers. However, some points should be addressed before publication.
<Major comments>
1. In Figure 2, is the values of 50 μM maximum inhibition? The authors should add the values of higher concentrations than 50 μM or that of treatment with the blocking antibody for integrin αvβ3 they used in Figure 3-6.
2. The authors show the results of the assays for cell migration and invasion in Figure 3 and 4. Incubation times of these experiments were 24h, 48h or 72h. In such long-time incubation, it is concerned about cell proliferation. The authors should mention how they inhibited cell proliferation in their experiments in the Materials and Methods section.
3. In Figure 4, why ψRGDechi-treated BT-549 cells exhibited a lower value of invasion than 1% FBS control, whereas the values of 1% FBS and ψRGDechi-treated MDA-MB-231 cells are similar? Does this mean that the sensitivity for ψRGDechi differ between these cell lines?
4. The method for quantification of loops is unclear in tube-formation assay. What size and shapes were quantified? What is the analyzed area, whole well?
5. In the section 2.7, the authors describe about vitronectin. Do BT-549 and MDA-MB-231 cells express and secrete vitronectin?
6. In Figure 7, why the authors focused on Akt? How about other signaling molecules?
Author Response
-Reviewer 1
Hill et al. inhibited integrin αvβ3 with ψRGDechi in mesenchymal subtype of triple-negative breast cancer (TNBC). The authors found higher integrin αv and β3 expressions in mesenchymal subtype of TNBC than the other TNBC subtypes. Treatment with ψRGDechi reduced adhesion, migration, invasion, formation of vascular-like structure and anchorage-independent growth in mesenchymal TNBC cell lines. Inhibition of αvβ3 with ψRGDechi reduced the expression levels of mesenchymal markers. However, some points should be addressed before publication.
We thank the Reviewer for giving us the opportunity to improve our work and we hope that he/she may appreciate the newly revised manuscript.
Major comments
1. In Figure 2, is the values of 50 μM maximum inhibition? The authors should add the values of higher concentrations than 50 μM or that of treatment with the blocking antibody for integrin αvβ3 they used in Figure 3-6.
Accordingly with Reviewer’s suggestion we tested the effect of the blocking antibody for integrin αvβ3 (LM609) on cell adhesion in both TNBC cell lines. These results are shown in new Figure 2 and reported in the Results Section (page 3, rows 117-118) and in the figure legend (page 4, rows 122-123) of Revised version of the manuscript.
2. The authors show the results of the assays for cell migration and invasion in Figure 3 and 4. Incubation times of these experiments were 24h, 48h or 72h. In such long-time incubation, it is concerned about cell proliferation. The authors should mention how they inhibited cell proliferation in their experiments in the Materials and Methods section.
We thank the Reviewer for his/her insightful and constructive comment which allows us to clarify this point. In according with the literature migration and invasion experiments using Boyden chamber were performed in presence of serum-free medium to minimize the influence of cell proliferation. Furthermore, following the Reviewer’s suggestion we performed MTS assay and found that ψRGDechi did not inhibit proliferation of TNBC cells at 24, 48 and 72 hours. These results are shown in the supplementary Figure S1 and reported in the Results section (page 4, rows 146-148) whereas MTS assay was described in Materials and Methods (page 12, rows from 335 to 343) of revised manuscript.
3. In Figure 4, why ψRGDechi-treated BT-549 cells exhibited a lower value of invasion than 1% FBS control, whereas the values of 1% FBS and ψRGDechi-treated MDA-MB-231 cells are similar? Does this mean that the sensitivity for ψRGDechi differ between these cell lines?
We perfectly agree with Reviewer, in fact, in all experiments performed BT-549 cells showed higher sensitivity for ψRGDechi than MDA-MB-231 cells. Probabily, we believe that this is due to different expression of mesenchymal markers such as vimentin and N-cadherin between these two cell lines. We comment this issue in the 2.8 Results section (page 8, rows from 231-232) of the Revised version of the manuscript.
4. The method for quantification of loops is unclear in tube-formation assay. What size and shapes were quantified? What is the analyzed area, whole well?
According to this Reviewer’s suggestion we clarified this point in further detail within the Materials and Methods section “Tube formation was analyzed under a phase-contrast microscopy and number of tubular structures identified as complete loops were quantified. Images were taken at 10x and average of the number of complete loops was calculated from 3-5 random fields by a macro made with ImageJ software” (page 13, rows 383-385) and in the legend of Figure 5 (page 7, rows 201-203) of revised manuscript (page 7).
5. In the section 2.7, the authors describe about vitronectin. Do BT-549 and MDA-MB-231 cells express and secrete vitronectin?
We thank the reviewer for highlighting this point and we agree that the sentence “Recently, it has been reported the involvement of vitronectin to drive stem cell differentiation through an integrin αvβ3-dependent mechanism [20]” can be misleading as reported in the results. Therefore, we removed it from section 2.7 and placed in the discussion of revised manuscript (pag10, rows 272-273) the following sentence “Recently, it has been reported that the expression of integrin αvβ3 in CSCs may confer not only tumorigenic features, but also metastatic capabilities through vitronectin engagement (Hurt et al Stem Cells 2010, 28: 390-8)”.
6. In Figure 7, why the authors focused on Akt? How about other signaling molecules?
We thank the reviewer for highlighting this point. We focused only on Akt signaling because many studies reported that this is one of the main pathways activated by αvβ3 integrin and correlated with EMT program (Luo J et al. J. Exp. Clin. Cancer Res. 2018, 37, 23; Luo BP et al. Curr. Med. Sci. 2018, 38, 467-472; Zhang PF et al. Cell Death Dis. 2016, 7, e2201).

Reviewer 2 Report
This article is written well, but lacks in new knowledge and grounds. Therefore, I require Major Revision.
・Please clarify the purpose of this study in “Introduction” section.
・Is the definition of MES - TNBC really accurate?
・Please clearly define the definition of CTC.
・This should be proved by gene expression profiles.
・Can clinically identify MES-TNBC easily? (such as immunohistochemistry)
・Please describe the background of ψ RGDechi in more detail in the text.
・With this study result alone, is not this conclusion too strong?
・Please describe the discussion in more detail.
Minor point
・The sentence of this paper has many careful mention errors. Please review it.
Author Response
-Reviewer 2
This article is written well, but lacks in new knowledge and grounds. Therefore, I require Major Revision.
We thank the Reviewer for giving us the opportunity to improve our work and we hope that he/she may appreciate the newly revised manuscript.
・Please clarify the purpose of this study in “Introduction” section.
We apologize for not having properly clarified the purpose in the introduction section. We did it in the revised manuscript and amended the introduction accordingly (page 2, rows from 80 to 85).
・Is the definition of MES - TNBC really accurate?
We thank the reviewer for highlighting this point. In the introduction section of revised manuscript (page 1-2, rows from 44 to 51) we addressed this issue and elucidated better the classification of TNBC subtypes performed by Lehmann and Burstein:
-Lehmann BD, Bauer JA, Chen X, Sanders ME, Chakravarthy AB, Shyr Y, et al. Identification of human triple-negative breast cancer subtypes and preclinical models for selection of targeted therapies. J Clin Invest. 2011, 121, 2750-67.
-Lehmann BD, Jovanović B, Chen X, Estrada MV, Johnson KN, Shyr Y et al. Refinement of Triple-Negative Breast Cancer Molecular Subtypes: Implications for Neoadjuvant Chemotherapy Selection. PLoS One. 2016, 11, e0157368.
-Burstein MD, Tsimelzon A, Poage GM, Covington KR, Contreras A, Fuqua SA et al. Comprehensive genomic analysis identifies novel subtypes and targets of triple-negative breast cancer. Clin Cancer Res. 2015, 21, 1688-98.
Furthermore, we performed the mRNA expression analysis of integrin subunits αV (ITGAV) and β3 (ITGB3) using the same dataset of 198 TNBC characterized through comprehensive genomic profiling by Burstein.
・Please clearly define the definition of CTC.
We believe that the Reviewer meant CSC (Cancer Stem Cells) instead CTC (Circulating Tumor Cells) because we did not address this topic in the manuscript. However, we explained in greater details the definition of CSC in the revised introduction (page 2, rows from 57 to 63).
・This should be proved by gene expression profiles.
In our manuscript, we observed that ψRGDechi was able to reduce the ability of MES-TNBC cells to form mammospheres that can be considered a cell population enriched for putative CSCs. We showed only that the peptide reduced levels of slug, a trascriptional repressor that is well known to be involved in EMT and cancer cell stemness. We are working on the effect of ψRGDechi on CSC gene expression profile, that we hope will constitute the basis of a future manuscript. Therefore, we prefer to change in the revised abstract the sentence “In addition, this peptide inhibited genes involved in EMT and CSC maintenance” in “In addition, this peptide reversed EMT program inhibiting mesenchymal markers” (page 1, rows from 29 to 30).
・Can clinically identify MES-TNBC easily? (such as immunohistochemistry)
Currently, unique to the mesenchymal TNBC subtype are patterns of gene expression associated with EMT and stemness. The immunohistochemical analysis and microarray assay of TNBC samples are based on markers associated with these processes. In this manuscript, we propose αvβ3 integrin, involved both in EMT and stemeness, as a potential biomarker of mesenchymal subgroup and we plan to validate it on TNBC samples in the close future.
・Please describe the background of ψ RGDechi in more detail in the text.
According to this Reviewer’s suggestion we describe ψRGDechi background in more detail in the introduction of revised manuscript (page 2, rows from 71 to 79).
・With this study result alone, is not this conclusion too strong?
According to this Reviewer’s suggestion we conclude with this sentence “However, further in vivo studies are needed in order to validate this novel peptide as a potential therapeutic agent in the treatment of this disease ” (page 13, rows from 411 to 413). Furthermore, in the abstract we reported that “These findings show that targeting αvβ3 integrin by ψRGDechi is possible to inhibit some of the malignant properties of MES-TNBC phenotype” (page 1, rows from 30-31).
・Please describe the discussion in more detail.
According to Reviewer’s criticism we have revised the discussion in more detail.
Minor point
・The sentence of this paper has many careful mention errors. Please review it.
We carefully proof-read the manuscript and eliminated mention errors.

Round 2
Reviewer 1 Report
No comment
Reviewer 2 Report
This paper has been revised and improved.